# Tunable single- and dual-wavelength lasers around 1.4 μm in Nd:LuGdAG crystal

**Haotian Huang, Yuzhao Li, Nguyen Tuan Anh, Yanfei Lü, Jing Xia** [ID]*

School of Physics and Astronomy, Yunnan University, Kunming, China

* xiajing_ynu@163.com

## Abstract

We present the first diode-pumped tunable single- and dual-wavelength (DW) laser operation near 1.4 μm spectral region in Nd:LuGdAG (Nd:LGAG) crystal on the $^4F_{3/2} \rightarrow {}^4I_{13/2}$ transition. Three distinct lasing wavelengths at 1414 nm, 1426 nm and 1437 nm were generated by adjusting a Lyot filter (LF) in the cavity, respectively. The maximum continuous-wave (CW) power output of 3.64 W at 1414 nm was attained under an absorbed pump power of 18.7 W, exhibiting a slope efficiency of 23.7% and optical conversion efficiency of 19.5%. Further, three pairs of DW lasers operating at 1414 nm and 1426 nm, 1414 nm and 1437 nm, 1426 nm and 1437 nm were also achieved, respectively. The DW operation at 1414 nm and 1437 nm yielded 2.82 W total CW output power, attaining 15.1% total optical conversion efficiency. Single- and DW lasers in the 1410–1440 nm spectral range have important application in fields such as optical communication and medicine.

## 1. Introduction

Nd-doped solid-state lasers predominantly employ three principal emission bands in the near-infrared spectrum: 0.9 μm corresponding to the three-level $^4F_{3/2} \rightarrow {}^4I_{9/2}$ transition, and four-level configurations at 1.1 μm ($^4F_{3/2} \rightarrow {}^4I_{11/2}$) and 1.3/1.4 μm ($^4F_{3/2} \rightarrow {}^4I_{13/2}$). Multiple Nd³⁺-doped crystals including Nd:YVO₄ [1,2], Nd:YAG [3–5], Nd:GdVO₄ [6,7], Nd:YLF [8,9], Nd:YAP [10,11], Nd:CALGO [12] and Nd:GSAG [13–15] have demonstrated solid-state laser functionality. Conventionally, the 1.3 μm emission band in Nd-doped crystals originates from the $^4F_{3/2} \rightarrow {}^4I_{13/2}$ transition. However, this splitting phenomenon arises from the crystal field splitting effect, which partitions the energy levels into multiple Stark sublevels. Exemplified by Nd:YAG, such crystal fields induce over a dozen distinct emission peaks within the $^4F_{3/2} \rightarrow {}^4I_{13/2}$ manifold, with corresponding 1.4 μm region emissions having been documented in multiple studies [16–19]. Laser sources operating near 1.4 μm exhibit inherent eye-safe characteristics, enabling their deployment in diverse technical domains including optical communications, coherent LIDAR systems, dermatological treatments, advanced laser medicine, and ophthalmic therapies [20–25]. Among neodymium-doped laser

**Data availability statement:** All relevant data are within the manuscript.

**Funding:** This work has been supported by the National Natural Science Foundation of China (Grant Nos. 62175209 and 62241506).

**Competing interests:** The authors have declared that no competing interests exist.

crystals, Nd:LGYAG has been widely adopted in solid-state lasers due to its excellent optical quality and weak thermal lensing effect [26,27]. In the case of Nd:LGAG, $Lu^{3+}$ and $Gd^{3+}$ ions substitute for the totality of the $Y^{3+}$ ions of the Nd:YAG but with a proportion $Lu^{3+}$ and $Gd^{3+}$ of about 30%. While The Nd:LGAG lasers at 1.1 [28], 0.9 [29] and 1.3 μm [30] have been implemented successfully in prior studies, systematic research on CW laser generation at 1.4 μm in the Nd:LGAG has not been reported until now. Fig 1 demonstrates the emission cross-section of the Nd:LGAG from 1250 nm to 1500 nm at room temperature on the $^4F_{3/2} \rightarrow {}^4I_{13/2}$ transition, which was calculated via the Füchtbauer-Ladenburg (F-L) formula [31]. It can be shown in Fig 1 that the strongest peak was 1332 nm. In addition, there were five peaks at 1316 nm, 1348 nm, 1414 nm, 1426 nm and 1437 nm.

In this study, we achieved three-wavelength tunability at 1414, 1426 and 1437 nm in Nd:LGAG on the $^4F_{3/2} \rightarrow {}^4I_{13/2}$ transition. Additionally, three pairs of DW tunability at 1414 nm and 1426 nm, 1414 nm and 1437 nm, 1426 nm and 1437 nm were also realized. Extensive implementation potential of DW laser systems has been identified across diverse technical domains including LIDAR systems [32], medical diagnostics [33], optical holography [34,35], precision spectral analysis [36], metrological sensing [37,38], nonlinear frequency conversion for UV/visible generation [39,40], and THz wave synthesis via difference frequency generation [41–43]. Especially, DW lasers around 1.4 μm enable simultaneous superficial epidermal heating and deep dermal collagen stimulation for non-invasive skin tightening and vascular coagulation, leveraging minimal thermal damage to surrounding tissues [44].

## 2. Experimental setup

The schematic diagram for the laser experiment was displayed in Fig 2. The pump system employs a 20 W 808 nm laser diode (LD) with a NA of 0.22 and a fiber core

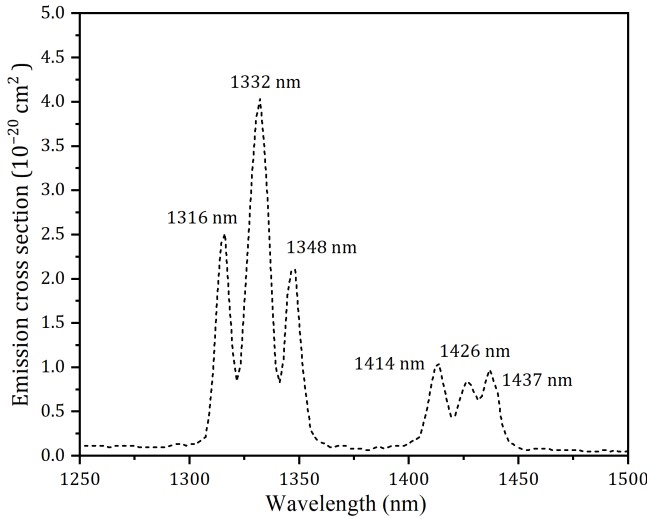

**Fig 1. Emission spectrum of the Nd:LGAG in 1250 −1500 nm.**

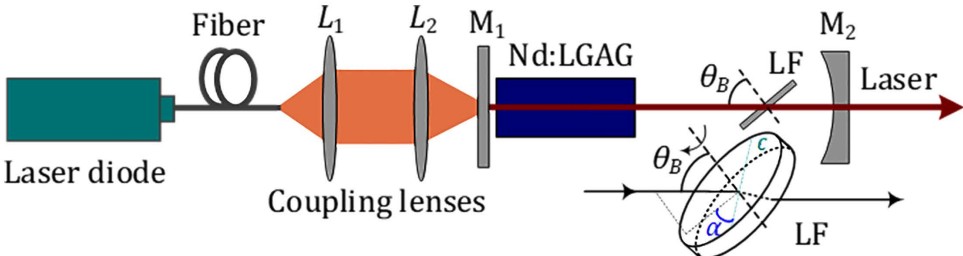

**Fig 2. Schematic setup for the laser experiment.** Inset: LF.

diameter of 400μm. The radius of the pump spot was about 200 μm in the active medium. The laser spot radius in the active medium was about 190 μm. Two identical coupling lenses ($L_1$ and $L_2$) with a focal length of 100 mm were utilized, featuring anti-reflection (AR) at 808 nm on both surfaces. The measured transmittance of the optical coupling system exceeded 98%. A Nd:LGAG (1.0 at.% doping, 6 mm length) functioned as the active medium with AR at 808 nm and 1410-1440 nm, which was sealed in indium foil and affixed to red copper mounts equipped with water cooling, maintained at 15°C.

The cavity input coupler was a planar mirror ($M_1$) with AR for 808 nm and 1060–1350 nm, and high reflectivity (HR) at 1410–1440 nm. The cavity output coupler was a concave mirror ($M_2$) with a radius of curvature of −200 mm, a transmittance ($T_{oc}$) of 3.5% at 1410–1440 nm, and AR at 1060–1350 nm. Two other couplers ($T_{oc}$ = 2.0% and 5.0%) were also carried out, with the $M_2$ demonstrated the optimal output performance. A quartz-based LF (4.0 mm thickness) was employed for wavelength tuning, positioned within the resonator at $\theta_B$ (Brewster angle) as depicted in Fig 2 inset. The tuning angle ($\alpha$) was an angle between the optical axis of the LF (C) and the incident light projection on the LF surface.

## 3. Results and discussion

The single-pass transmittance for different wavelengths transmitted through the LF was expressed as [45]

$$T_{Lyot,i} = 1 - \frac{4cos^2\alpha sin^2\theta_B}{1-4cos^2\alpha cos^2\theta_B}\left(1 - \frac{cos^2\alpha sin^2\theta_B}{1-cos^2\alpha cos^2\theta_B}\right)sin^2\left(\frac{\delta_i}{2}\right),$$

(1)

where $i$ = 1, 2 and 3 represents the 1414 nm, 1426 nm and 1437 nm three wavelengths, respectively, $\delta_i = 2\pi d(n_o - n_e)(1 - cos^2\alpha cos^2\theta_B)/\lambda_i sin\theta_B$ is an optical phase difference, $n_o$ and $n_e$ are the refractive indices of o- and e-light, respectively, $d$ is a thickness of the filter. With Eq. (1) and the parameters: $n_o$ = 1.5443, $n_e$ = 1.5534, $d$ = 4 mm and $\theta_B$ = 57.2°, the round-trip transmittance($T^2_{Lyot,i}$) of the different laser wavelength ($\lambda_i$) was calculated as a function of the tuning angle as displayed in Fig 3. It can be shown from Fig 3 that $T^2_{Lyot,i}$ can be controlled by regulating the LF surface around its normal axis. Thus, tuning between emission wavelengths can be realized by controlling the LF.

When tuning angle was rotated to about 38°, 39°, and 40°, the corresponding emissions at 1414 nm, 1426 nm, and 1437 nm were achieved, and their output-input performances were displayed in Fig 4. At an absorbed pump power of 18.7 W (corresponding to an incident power of 20 W) with $T_{oc}$ = 3.5%, the laser demonstrated output powers of 3.64 W at 1414 nm, 3.01 W at 1426 nm, and 2.21 W at 1437 nm. The corresponding lasing thresholds were 2.70 W, 3.01 W and 3.21 W, with slope efficiencies of 23.7%, 19.4% and 14.5%, respectively. At $T_{oc}$ = 2.0%, the laser demonstrated slope efficiencies of 18.3%, 15.2%, and 10.3% at 1414 nm, 1426 nm, and 1437 nm respectively, with corresponding threshold powers of 1.72 W, 2.12 W and 2.65 W. When $T_{oc}$ was increased to 5.0%, the slope efficiencies changed to 17.4%, 14.6%, and 8.5%, while the threshold powers increased to 4.5 W, 5.7 W and 6.3 W for the respective wavelengths. The laser spectra

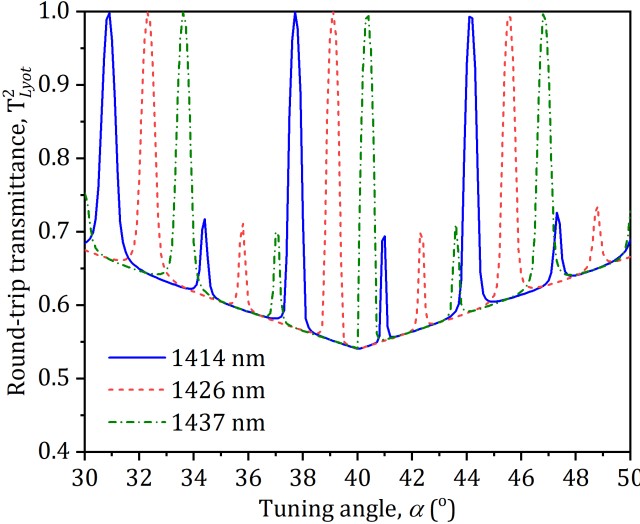

**Fig 3. Round-trip transmittance ($T2$ *Lyot,i*) versus the tuning angle (α).**

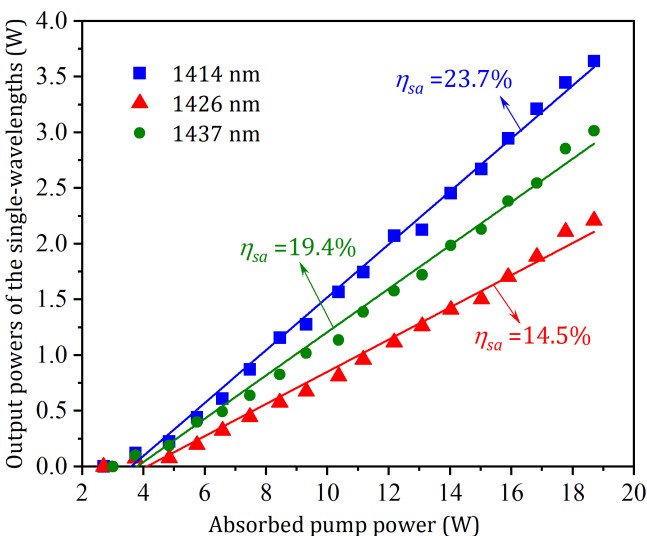

**Fig 4. Output powers of the three single-wavelengths versus absorbed pump power.**

at 1414 nm, 1426 nm, and 1437 nm at the maximum pumping were displayed in the Fig 5. The corresponding wavelength peaks (1415.85 nm, 1426.28 nm and 1436.94 nm) exhibited spectral line width (FWHM) values of 0.30 nm, 0.33 nm and 0.35 nm, respectively.

The power stabilities of the three laser wavelengths were measured with a precision power meter. The power fluctuations (RMS) at the maximum output powers were about 2.7%, 3.6% and 3.9% in 1 hour, respectively, as shown in Fig 6. The insets (a)-(c) of Fig 6 show the measured radii and the beam quality factors ($M^2$) of the 1414 nm, 1426 nm, and 1437 nm beams, respectively. The beam quality factors ($M^2$) of the 1414 nm, 1426 nm, and 1437 nm wavelengths were

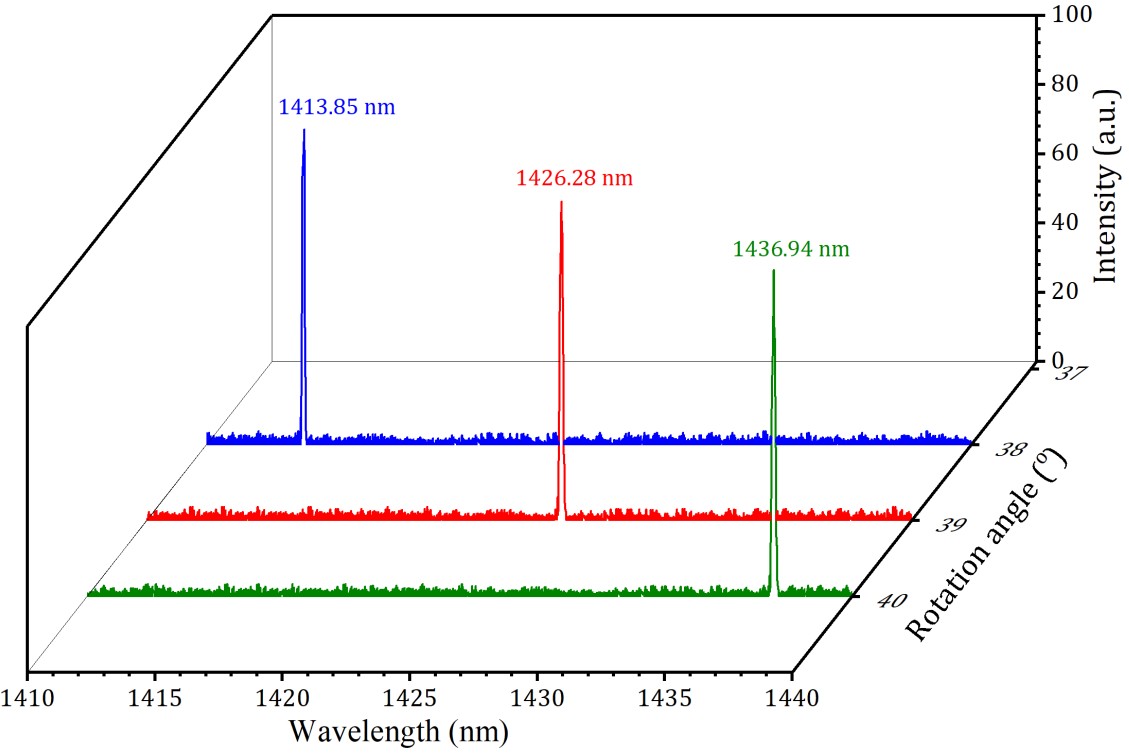

**Fig 5. Laser spectra of the three single-wavelengths.**

measured using the knife-edge technique. The corresponding values in both transverse directions at maximum output power were less than 1.16, 1.12 and 1.25, respectively.

For a four-level laser system operating with CW, the oscillation threshold of each emission wavelength in a DW operation was given by [46]

$$P_{tha,i} = \frac{-\ln{(1-T_{oc})} + L_i + L_{0i}}{2l_c\eta_{q,i}} \frac{h\nu_p}{\sigma_i\tau_i} \frac{1}{\iiint r_p(r,z)s_i(r,z)d\upsilon},$$ (2)

where $T_{oc}$ is the cavity transmittance for the laser emission wavelengths, $L_i = 1 - T^2_{Lyot,i}$ is the round-trip loss, which is caused by the LF, $L_{0i}$ is the cavity round trip passive loss, $l_c$ is the length of the Nd:LGAG, $\eta_{q',i}$ is the quantum efficiency, $h\nu_p$ is the photon energy of the pump beam, $\sigma_i$ is the stimulated emission cross-section, $\tau_i$ is the upper energy level lifetime, $r_p(r,z)$ is the pump beam distribution of the normalized intensity in the Nd:LGAG, and $s_i(r,z)$ is the cavity mode distribution of the normalized intensity for the emission wavelength. $r_p(r,z)$ and $s_i(r,z)$ can be written by [47], respectively,

$$r_p(r,z) = \frac{\alpha e^{-\alpha z}}{\pi \omega^2_p(z)(1-e^{-\alpha z})} \Theta\left(\omega^2_p(z) - r^2\right),$$ (3)

and

$$s_i(r,z) = \frac{2}{\pi \omega^2_{i0} l_c} exp\left(\frac{2r^2}{\omega^2_{i0}}\right),$$ (4)

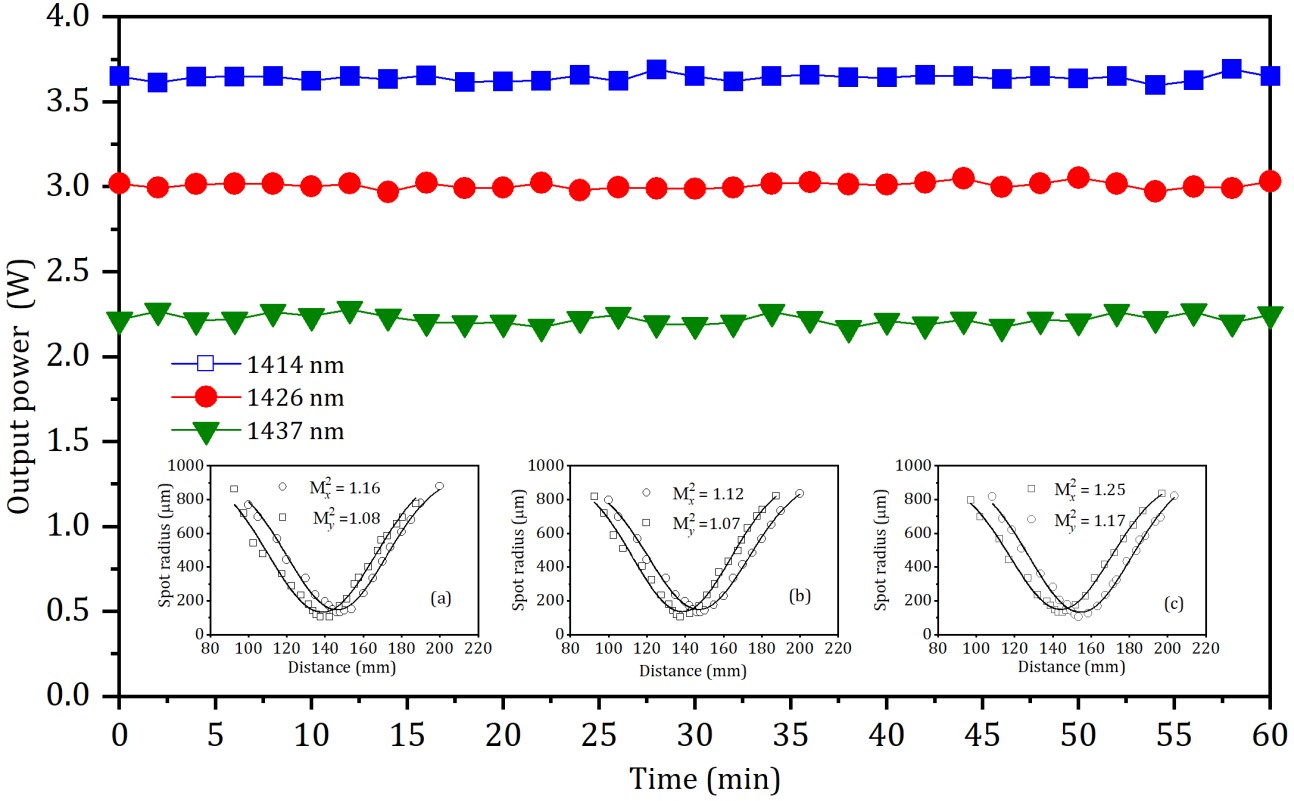

**Fig 6. Power stabilities of the three laser wavelengths.** Insets (a), (b) and (c) show the X- and Y-axes radii as functions of Z-axis position for the 1414 nm, 1426 nm and 1437 nm beams, respectively.

where $\alpha$ is the absorption coefficient, $\Theta$ is the Heaviside step function, $\omega_{0i}$ is the radius of the laser spot, and the size of the pump beam in the Nd:LGAG given by

$$\omega_p^2(z) = \omega_{p0}^2 \left\{ 1 + \left[ \frac{\lambda_p M_p^2}{n \pi \omega_{p0}^2} (z - z_0) \right]^2 \right\},$$

(5)

where $\omega_{p0}$ is the radius of the pump spot, $\lambda_p$ is the pump wavelength, $M2$ $p$ is the quality factor of the pump beam. With Eqs. (1)-(5) and the parameters in our experiment: $T_{oc} = 3.5\%$, $\sigma_1 = 1.04 \times 10^{-20}$ cm², $\sigma_2 = 0.84 \times 10^{-20}$ cm², $\sigma_3 = 0.98 \times 10^{-20}$ cm², $\alpha = 3.5$ cm⁻¹, $l_c = 5$ mm, $\omega_p = 200$ µm, $\omega_{0i} = 190$ µm, $n = 1.83$, $M^2 = 3.5$, $\eta_{q,i} \approx 0.57$, $\tau_i \approx 262$ µs, $hv_p = 2.45 \times 10^{-19}$ J, and $L_{0i} = 0.5\%$ was measured using the Findlay-Clay method [48], the threshold was calculated as a function of tuning angle $\alpha$ for the three laser wavelengths, as displayed in Fig 7. It can be seen that the threshold can be controlled by regulating the LF. It can be observed that intersecting points exist between any two threshold power curves, indicating that the pump power required for lasing threshold is identical for both wavelengths at these intersection points. Consequently, DW lasers could be achieved when the pump powers were precisely regulated to these power levels.

When $\alpha$ was regulated to about 3.5°, 5.5°, and 16.5°, the three pairs of the DWs at 1414 nm and 1426 nm, 1414 nm and 1437 nm, and 1426 nm and 1437 nm were generated, respectively, and their output-input performances were displayed in Fig 8. At an absorbed power of 18.7 W, the total powers were 2.82 W (1.44 W at 1437 nm and 1.38 W at 1414 nm), 2.75

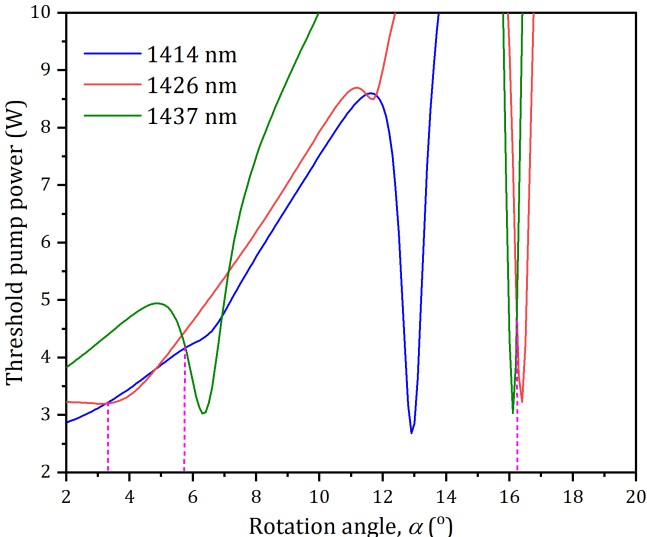

**Fig 7. Tuning angle versus the threshold power.**

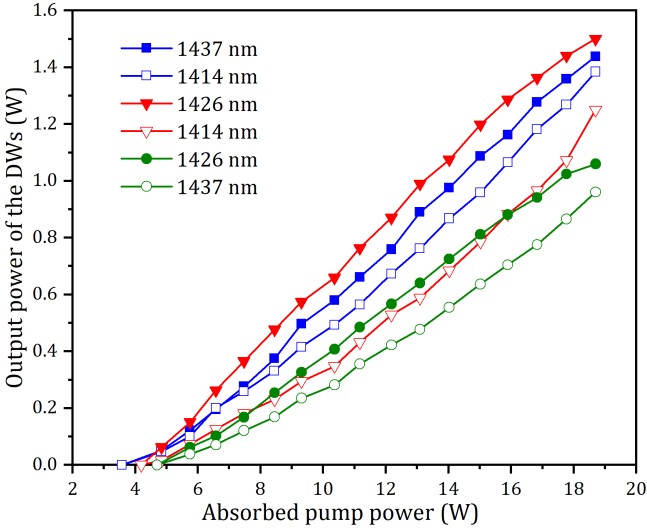

**Fig 8. Output powers of DWs versus absorbed pump power.**

W (1.50 W at 1426 nm and 1.25 W at 1414 nm) and 2.12 W (1.06 W at 1426 nm and 0.96 W at 1437 nm) for the three pairs of DWs, respectively. The corresponding threshold powers were 3.58 W, 4.21 W and 4.72 W, respectively. The total optical conversion efficiencies with respect to the absorbed power were 15.1%, 14.7% and 11.3%, respectively. The laser spectra of the three pairs of DWs were displayed in the Fig 9. The corresponding peak wavelengths were 1413.83 nm and 1426.41 nm, 1413.95 nm and 1437.04 nm, 1426.44 nm and 1437.02 nm, respectively. For the three pairs of DWs, Their corresponding $M^2$ factors were 1.12 and 1.15, 1.18 and 1.24 and 1.22 and 1.27, respectively, and their power stabilities were about 2.5% and 2.9%, 2.8% and 3.8%, and 3.5% and 4.2%, respectively.

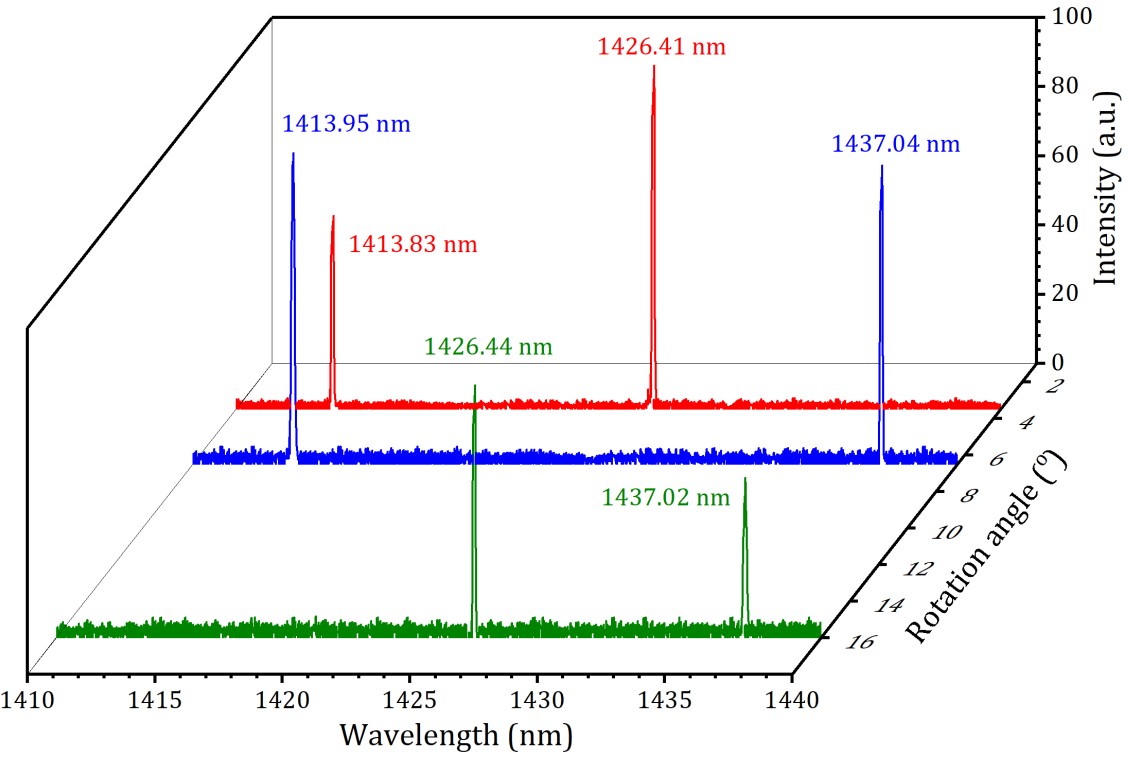

**Fig 9. Laser spectra of the three pairs of DWs.**

Compared with the previously reported the 1.4 μm single-wavelength laser on Nd:GSAG crystal (slope efficiency of 13.6%, optical conversion efficiency of 11.5% [15]), the single-wavelength system in this study achieved a slope efficiency of 23.7% and an optical conversion efficiency of 19.5%. In terms of DW laser output, the total optical conversion efficiency has increased from 9.2% to 15.1%. These data fully demonstrate the significant progress of this laser system in the optical conversion efficiency at the 1.4 μm spectral region.

## 4. Conclusion

In conclusion, diode-pumped tunable single- and DW laser operation near 1.4 μm spectral region in Nd:LGAG on the $^4F_{3/2} \rightarrow ^4I_{13/2}$ transition was demonstrated for the first time. By regulated an intracavity LF, the three single-wavelengths at 1414 nm, 1426 nm and 1437 nm were obtained, respectively. The maximum CWoutput power of 3.64 W at 1414 nm was attained under an absorbed pump power of 18.7 W, exhibiting a slope efficiency of 23.7% and optical conversion efficiency of 19.5%. Further, three pairs of DW lasers operating at 1414 nm and 1426 nm, 1414 nm and 1437 nm, and 1426 nm and 1437 nm were also achieved, respectively. The DW operation at 1414 nm and 1437 nm yielded 2.82 W total CW output power, attaining 15.1% total optical conversion efficiency. This study proposes a new method for generating tunable single- and DW lasers, which can be applied to other active medium to achieve laser output of different spectral regions.

## Author contributions

**Data curation:** Haotian Huang, Yuzhao Li, Nguyen Tuan Anh, Jing Xia.

**Formal analysis:** Yuzhao Li.

**Investigation:** Yuzhao Li, Nguyen Tuan Anh.

**Project administration:** Yanfei Lü, Jing Xia.

**Writing – original draft:** Haotian Huang, Yuzhao Li, Yanfei Lü, Jing Xia.

**Writing – review & editing:** Yanfei Lü, Jing Xia.

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
