## [Decision Letter · Decision Letter 0]

25 Jul 2025

Dear Dr. Xia,

Thank you for submitting your manuscript to PLOS ONE. After careful consideration, we feel that it has merit but does not fully meet PLOS ONE’s publication criteria as it currently stands. Therefore, we invite you to submit a revised version of the manuscript that addresses the points raised during the review process.

We look forward to receiving your revised manuscript.

Kind regards,

Rajesh Sharma

Academic Editor

PLOS ONE

Journal Requirements: 

 [This work has been supported by the National Natural Science Foundation of China (Grant Nos. 62175209 and 62241506).]. 

[This work has been supported by the National Natural Science Foundation of China (Grant Nos. 62175209 and 62241506).]

[This work has been supported by the National Natural Science Foundation of China (Grant Nos. 62175209 and 62241506).]. 

Reviewers' comments:

Reviewer's Responses to Questions

**Comments to the Author**

1. Is the manuscript technically sound, and do the data support the conclusions?

Reviewer #1: Yes

Reviewer #2: Yes

Reviewer #3: Yes

Reviewer #4: No

2. Has the statistical analysis been performed appropriately and rigorously?

Reviewer #1: Yes

Reviewer #2: Yes

Reviewer #3: Yes

Reviewer #4: No

3. Have the authors made all data underlying the findings in their manuscript fully available?

Reviewer #1: Yes

Reviewer #2: Yes

Reviewer #3: No

Reviewer #4: No

4. Is the manuscript presented in an intelligible fashion and written in standard English?

Reviewer #1: Yes

Reviewer #2: Yes

Reviewer #3: Yes

Reviewer #4: No

Reviewer #1: Comments

The authors report a laser output at 1.4 μm is achieved in the Nd:LuGdAG crystal, corresponding maximum output power is 3.64 W with the slope efficiency of 23.7%. By tuning the angle of the Lyot filter, three distinct lasing wavelengths at 1414, 1426 and 1437 nm laser wavelengths were obtained. Besides, three pairs of dual-wavelength lasers operating at 1414 and 1426 nm, 1414 and 1437 nm, 1426 and 1437 nm were also achieved, respectively. The manuscript can be accepted for publication after some revisions. The suggestions are as follows:

1. What is the output wavelength and output power of Nd:LuGdAG laser without Lyot filter inserted into the resonant cavity?

2. What are the losses of the coupled lenses (L1 and L2)?

3. “The Nd:LGAG lasers at 1.1 μm on the 4F3/2→4I11/2 [28], 0.9 μm on the 4F3/2→4I9/2 [29] and 1.3 μm on the 4F3/2→4I13/2 [30] transitions have been reported. To the best of our knowledge, CW lasers around 1.4 μm in Nd:LGAG have not been studied until now.” This sentence should be deleted. This sentence should be deleted because it is repetitive with the following sentence.

4. The author should provide the measurement method of the beam quality factor.

5. What does the parameter “Θ” in Eq. 3 represent? It is advisable for authors to express clearly it in the manuscript.

Reviewer #2: The manuscript by Huang et al. reports on a tunable single- and dual-wavelength lasers around 1.4 μm in Nd:LuGdAG crystal. The authors demonstrate the wavelength tunability of the output and report on the maximum output power of 3.64 W, corresponding to 23.7% optical efficiency. In addition, three pairs of dual-wavelength lasers operating at 1.4 μm were also achieved. The results are novel, and the manuscript is well-written. The details of the experiment are provided adequately. I recommend the publication of the manuscript after the following corrections:

1. How did the author measure the output power at these close dual wavelengths separately?

2. In the introduction: “The Nd:LGAG lasers at 1.1 μm on the 4F3/2→4I11/2 [28], 0.9 μm on the 4F3/2→4I9/2 [29] and 1.3 μm on the 4F3/2→4I13/2 [30] transitions have been reported. To the best of our knowledge, CW lasers around 1.4 μm in Nd:LGAG have not been studied until now. While The Nd:LGAG lasers at 1.1 [28], 0.9 [29] and 1.3 μm [30] have been implemented successfully in prior studies, systematic research on CW laser generation at 1.4 μm in the Nd:LGAG has not been reported until now.”

These two sentences are repetitive. One of them should be deleted.

3. In the experimental setup: “A Nd:LGAG (1.0 at.% doping, 6 mm length) functioned as the active medium…” To my knowledge, the output parameters of the diode-pumped laser are related to the length and doping concentration of the laser crystal. Generally, the doping concentration of the gain medium in pure three-level lasers is very low to reduce thermal effects and reabsorption. How is the doping concentration of 1.0 at. % chosen?

4. What is the cavity mode radius? How is it determined, under considering the thermal lens effect?

5. How to measure the beam quality factor?

Reviewer #3: This paper presents a study on diode-pumped tunable single- and dual-wavelength laser operation in Nd:LGAG crystal. The research achieve the single- and dual-wavelength laser operation near the 1.4 μm spectral region in Nd:LGAG crystal, it has certain innovation and value.

The paper can better reflect the quality and significance of the research with appropriate supplements and improvements.

1. In Page 3” The cavity output coupler was a concave mirror (M2) with a radius of curvature of –200 mm, a transmittance (Toc) of 3.5% at 1410-1440 nm, and AR at 1060-1350 nm. Two other couplers (Toc = 3.5% and = 5.0%) were also carried out, with the M2 demonstrated the optimal output performance.” The other couplers of 3.5% is contradictory to the previous. And it is suggested to show the output powers at different transmittance.

2. It is excellent that the M2 factors of the 1414, 1426 and 1437 nm wavelengths were 1.16, 1.12 and 1.25 at the maximum output powers. Suggested add the test figure.

Reviewer #4: The work by Huang et al. investigated the diode-pumped tunable single- and dual-wavelength (DW) laser operation near 1.4 μm spectral region in Nd:LuGdAG (Nd:LGAG) crystal. The work may be of some significance for the development of solid-state lasers. But there remain many issues. I can’t recommend the acceptance till all my concerns below are well responded:

1.In the Introduction section, Fig. 1 displayed the emission spectrum of the Nd:LGAG in 1250 -1500 nm. Is it the result of the manuscript or the result of other literatures? And why do the authors include pictures in the Introduction section?

2.Fig. 5 displayed the laser spectra of the three single-wavelength lasers. Please provide the corresponding spectra at different pump powers.

3.The authors stated “Their corresponding M2 factors were 1.12 and 1.15, 1.18 and 1.24 and 1.22 and 1.27, respectively, and their power stabilities were about 2.5 and 2.9%, 2.8 and 3.8% and 3.5 and 4.2%, respectively”. The authors should prove the M2 factors and power stabilities by providing the corresponding pictures.

4.The authors can compare the performance of different lasers at near 1.4 μm to visually show the advantages of this work.

5.The English is casual at some places. Please check and revise the manuscript thoroughly and carefully.

**Do you want your identity to be public for this peer review?** For information about this choice, including consent withdrawal, please see our Privacy Policy

Reviewer #1: No

Reviewer #2: No

Reviewer #3: No

Reviewer #4: No

---

## [Author Response · Author response to Decision Letter 1]

15 Aug 2025

Comments

Reviewer #1: The authors report a laser output at 1.4 μm is achieved in the Nd:LuGdAG crystal, corresponding maximum output power is 3.64 W with the slope efficiency of 23.7%. By tuning the angle of the Lyot filter, three distinct lasing wavelengths at 1414, 1426 and 1437 nm laser wavelengths were obtained. Besides, three pairs of dual-wavelength lasers operating at 1414 and 1426 nm, 1414 and 1437 nm, 1426 and 1437 nm were also achieved, respectively. The manuscript can be accepted for publication after some revisions. The suggestions are as follows:

1. What is the output wavelength and output power of Nd:LuGdAG laser without Lyot filter inserted into the resonant cavity?

Response: In fact, the Lyot filter has no loss to the operating wavelength (because the transmittance is 100% at the operating wavelength), it only suppresses other wavelengths besides the operating wavelength.

2. What are the losses of the coupled lenses (L1 and L2)?

Response: Two identical coupling lenses (L1 and L2) were antireflection coated at 808 nm. The total transmittance the lenses is greater than 98%, so we ignore the loss of the lenses. We describe the coupling efficiency of the two lenses in the revised version.

3. “The Nd:LGAG lasers at 1.1 μm on the 4F3/2→4I11/2 [28], 0.9 μm on the 4F3/2→4I9/2 [29] and 1.3 μm on the 4F3/2→4I13/2 [30] transitions have been reported. To the best of our knowledge, CW lasers around 1.4 μm in Nd:LGAG have not been studied until now.” This sentence should be deleted. This sentence should be deleted because it is repetitive with the following sentence.

Response: We have deleted this repetitive sentence in the revised version.

4. The author should provide the measurement method of the beam quality factor.

Response: We have added the measurement method for beam quality factor in the revised version.

5. What does the parameter “Θ” in Eq. 3 represent? It is advisable for authors to express clearly it in the manuscript.

Response: Θ is the Heaviside step function, and we have explained it in the revised version.

Reviewer #2: The manuscript by Huang et al. reports on a tunable single- and dual-wavelength lasers around 1.4 μm in Nd:LuGdAG crystal. The authors demonstrate the wavelength tunability of the output and report on the maximum output power of 3.64 W, corresponding to 23.7% optical efficiency. In addition, three pairs of dual-wavelength lasers operating at 1.4 μm were also achieved. The results are novel, and the manuscript is well-written. The details of the experiment are provided adequately. I recommend the publication of the manuscript after the following corrections:

1. How did the author measure the output power at these close dual wavelengths separately?

Response: The 1.4 μm laser beam was separated using the flexible wavelength selector, and the output power of each wavelength was measured respectively.

2. In the introduction: “The Nd:LGAG lasers at 1.1 μm on the 4F3/2→4I11/2 [28], 0.9 μm on the 4F3/2→4I9/2 [29] and 1.3 μm on the 4F3/2→4I13/2 [30] transitions have been reported. To the best of our knowledge, CW lasers around 1.4 μm in Nd:LGAG have not been studied until now. While The Nd:LGAG lasers at 1.1 [28], 0.9 [29] and 1.3 μm [30] have been implemented successfully in prior studies, systematic research on CW laser generation at 1.4 μm in the Nd:LGAG has not been reported until now.”

These two sentences are repetitive. One of them should be deleted.

Response: We have deleted this repetitive sentence in the revised version.

3. In the experimental setup: “A Nd:LGAG (1.0 at.% doping, 6 mm length) functioned as the active medium…” To my knowledge, the output parameters of the diode-pumped laser are related to the length and doping concentration of the laser crystal. Generally, the doping concentration of the gain medium in pure three-level lasers is very low to reduce thermal effects and reabsorption. How is the doping concentration of 1.0 at. % chosen?

Response: Usually, low doping reduces the thermal effect of crystals. However, the absorption efficiency of Nd:LGAG crystals is relatively low. Therefore, while ensuring high absorption efficiency, the doping concentration should be reduced as much as possible. We used 1.0% doping and a 6 mm long Nd:LGAG crystal, but the absorption efficiency of the crystal was 93.5%.

4. What is the cavity mode radius? How is it determined, under considering the thermal lens effect?

Response: The laser spot radius in the active medium was about 190 μm, which is calculated based on the ABCD matrix considering the crystal thermal lens.

5. How to measure the beam quality factor?

Response: We measured the M2 factor using the knife-edge technique. We have added it in the revised version.

Reviewer #3: This paper presents a study on diode-pumped tunable single- and dual-wavelength laser operation in Nd:LGAG crystal. The research achieve the single- and dual-wavelength laser operation near the 1.4 μm spectral region in Nd:LGAG crystal, it has certain innovation and value.

The paper can better reflect the quality and significance of the research with appropriate supplements and improvements.

Response: Thank you for your constructive comments on this study. There is no doubt that these professional suggestions provide important directions for improving the quality of papers.

1. In Page 3” The cavity output coupler was a concave mirror (M2) with a radius of curvature of –200 mm, a transmittance (Toc) of 3.5% at 1410-1440 nm, and AR at 1060-1350 nm. Two other couplers (Toc = 3.5% and = 5.0%) were also carried out, with the M2 demonstrated the optimal output performance.” The other couplers of 3.5% is contradictory to the previous. And it is suggested to show the output powers at different transmittance.

Response: Toc = 3.5% and = 5.0% should be Toc = 2.0% and = 5.0% respectively. We have corrected it in the revised version. To more clearly compare the slope efficiency of different emission wavelengths, we present the test results of three wavelengths in the same graph. Considering that displaying three curves with different transmittance simultaneously would lead to overly dense graphical information, we have supplemented the slope efficiency and threshold data of the other two wavelengths in the "Results and Discussion" section to ensure the complete presentation of the experimental data. This processing approach not only maintains the simplicity of the illustration but also fully presents all the key experimental data.

2. It is excellent that the M2 factors of the 1414, 1426 and 1437 nm wavelengths were 1.16, 1.12 and 1.25 at the maximum output powers. Suggested add the test figure.

Response: We measured the M2 factor using the knife-edge technique. The value of the M2 factor is calculated by the measured beam waist radius and beam divergence angle. We have supplemented the measurement method and corresponding measured data for beam quality in the revised version.

Reviewer #4: The work by Huang et al. investigated the diode-pumped tunable single- and dual-wavelength (DW) laser operation near 1.4 μm spectral region in Nd:LuGdAG (Nd:LGAG) crystal. The work may be of some significance for the development of solid-state lasers. But there remain many issues. I can’t recommend the acceptance till all my concerns below are well responded:

Response: We thank you very much for the kind consideration and constructive comments. All your comments are very professional, pertinent and kind, and will be of great help to the improvement of our manuscript. We have carefully revised the manuscript and provided the point-by-point response below in blue. The changes in the revised manuscript have been highlighted.

1. In the Introduction section, Fig. 1 displayed the emission spectrum of the Nd:LGAG in 1250 -1500 nm. Is it the result of the manuscript or the result of other literatures? And why do the authors include pictures in the Introduction section?

Response: The data in Figure 1 is calculated through Füchtbauer-Ladenburg formula. Figure 1 is introduced in the introduction section to show the emission peak of the Nd:LGAG crystal in the 1410-1440 nm band. The results of this spectral characteristic analysis naturally extend to the key working bands that this study intends to achieve.

2. Fig. 5 displayed the laser spectra of the three single-wavelength lasers. Please provide the corresponding spectra at different pump powers.

Response: In fact, the emission wavelength of the crystal does not drift with the change of pump power, so the emission spectrum is the same at any pump power.

3. The authors stated “Their corresponding M2 factors were 1.12 and 1.15, 1.18 and 1.24 and 1.22 and 1.27, respectively, and their power stabilities were about 2.5 and 2.9%, 2.8 and 3.8% and 3.5 and 4.2%, respectively”. The authors should prove the M2 factors and power stabilities by providing the corresponding pictures.

Response: We measured the M2 factor using the knife-edge technique. The value of the M2 factor is calculated by the measured beam waist radius and beam divergence angle. We have supplemented the corresponding measured data (Insets (a-c) of Fig.6) for beam quality in the revised version. In addition, we have also supplemented the measured pictures of the power stability in the revised version (Fig.6).

4. The authors can compare the performance of different lasers at near 1.4 μm to visually show the advantages of this work.

Response: This is a very good suggestion. We have added a comparison with a 1.4 μm wavelength laser at the end of the results and discussion. This can indeed visually demonstrate the advantages of this work.

5. The English is casual at some places. Please check and revise the manuscript thoroughly and carefully.

Response: We have conducted a comprehensive review and optimization of the English expression throughout the text to ensure that the language is smooth and natural, and the expression is accurate and standardized.

---

## [Decision Letter · Decision Letter 1]

15 Sep 2025

Tunable single- and dual-wavelength lasers around 1.4 μm in Nd:LuGdAG crystal

PONE-D-25-26624R1

Dear Dr. Xia,

We’re pleased to inform you that your manuscript has been judged scientifically suitable for publication and will be formally accepted for publication once it meets all outstanding technical requirements.

Kind regards,

Rajesh Sharma

Academic Editor

PLOS ONE

Additional Editor Comments (optional):

Reviewer #1:

Reviewer #2:

Reviewer #4:

Reviewers' comments:

Reviewer's Responses to Questions

**Comments to the Author**

Reviewer #1: All comments have been addressed

Reviewer #2: All comments have been addressed

Reviewer #4: (No Response)

2. Is the manuscript technically sound, and do the data support the conclusions?

Reviewer #1: Yes

Reviewer #2: Yes

Reviewer #4: (No Response)

3. Has the statistical analysis been performed appropriately and rigorously?

Reviewer #1: Yes

Reviewer #2: Yes

Reviewer #4: (No Response)

4. Have the authors made all data underlying the findings in their manuscript fully available?

Reviewer #1: Yes

Reviewer #2: Yes

Reviewer #4: (No Response)

5. Is the manuscript presented in an intelligible fashion and written in standard English?

Reviewer #1: Yes

Reviewer #2: Yes

Reviewer #4: (No Response)

Reviewer #1: The authors have fully resolved all the issues I raised in their revision. Therefore, I recommend that the manuscript be accepted for publication in its current form.

Reviewer #2: The queries raised have been addressed satisfactorily by the authors, and I can recommend the publication of this article.

Reviewer #4: In the manuscript, the author has made revisions according to my review comments, and the paper can be accepted.

**Do you want your identity to be public for this peer review?** For information about this choice, including consent withdrawal, please see our Privacy Policy

Reviewer #1: No

Reviewer #2: No

Reviewer #4: No

---

## [Editor Report · Acceptance letter]

PONE-D-25-26624R1

PLOS ONE

Dear Dr. Xia,

I'm pleased to inform you that your manuscript has been deemed suitable for publication in PLOS ONE. Congratulations! Your manuscript is now being handed over to our production team.

Kind regards,

on behalf of

Dr. Rajesh Sharma

Academic Editor

PLOS ONE